# Design and Study of Machine Tools for the Fly-Cutting of Ceramic-Copper Substrates

**DOI:** 10.3390/ma17051111

**Published:** 2024-02-28

**Authors:** Chupeng Zhang, Jiazheng Sun, Jia Zhou, Xiao Chen

**Affiliations:** School of Mechanical Engineering, Hubei University of Technology, Wuhan 430068, China; 20181048@hbut.edu.cn (C.Z.); 102110079@hbut.edu.cn (J.S.); 15272757302@139.com (J.Z.)

**Keywords:** ceramic-copper substrate, fly-cutting, roughness

## Abstract

Ceramic-copper substrates, as high-power, load-bearing components, are widely used in new energy vehicles, electric locomotives, high-energy lasers, integrated circuits, and other fields. The service length will depend on the substrate’s copper-coated surface quality, which frequently achieved by utilising an abrasive strip polishing procedure on the substrate’s copper-coated surface. Precision diamond fly-cutting processing machine tools were made because of the low processing accuracy and inability to match the production line’s efficiency. An analysis of the fly-cutting machining principle and the structural makeup of the ceramic-copper substrate is the first step in creating a roughness prediction model based on a tool tip trajectory. This model demonstrates that a shift in the tool tip trajectory due to spindle runout error directly impacts the machined surface’s roughness. The device’s structural optimisation design is derived from the above analyses and implemented using finite element software. Modal and harmonic response analysis validated the machine’s gantry symmetrical structural layout, a parametric variable optimisation design optimised the machine tool’s overall dimensions, and simulation validated the fly-cutterring’s constituent parts. Enhancing the machine tool’s stability and motion accuracy requires using the LK-G5000 laser sensor to measure the guideway’s straightness. The result verified the machine tool’s design index, with the Z- and Y-axes’ straightness being better than 2.42 μm/800 mm and 2.32 μm/200 mm, respectively. Ultimately, the device’s machining accuracy was confirmed. Experiments with flying-cut machining on a 190 × 140 mm ceramic-copper substrate yielded a roughness of Sa9.058 nm. According to the experimental results, the developed machine tool can fulfil the design specifications.

## 1. Introduction

Electric locomotives, green cars, integrated circuits, and high-energy lasers are just a few of the quickly developing industries that extensively use heat-resistant, ceramic-copper substrates. These substrates boast exceptional thermal conductivity and power efficiency, which means their surface quality directly impacts their longevity [1,2,3,4]. Currently, surface of copper cladding is usually processed by abrasive belt grinding, which is time-consuming and less accurate. Moreover, its requires higher levels of accuracy and efficiency to meet production line demands. It might be beneficial to develop a type of diamond precision fly-cutting machine tool that can be applied to the ceramic-copper substrate because this type of ultraprecision and precision machining technology can realise the rapid removal of surface materials on metal and ceramics and effectively improve the machining surface finish and machining accuracy [5,6,7,8].

The design of precision machine tools has been thoroughly researched by academics from around the world. A typical precision machining technology is the diamond precision fly-cutting machining process. In the design and optimisation of precision machines, to reduce the impact of dynamic errors within the machine on the stability of the device and the accuracy of the machining, the static and dynamic characteristics and the machine’s thermal deformation must be analysed [9,10,11,12,13]. The use of a gantry symmetrical structure in the design of precision machine tools can effectively reduce the impact of thermal deformation on the device; in order to ensure its performance, the machine tool should be considered, first of all, with reference to the effect of gravity on its static deformation [14,15]. In research to improve the machining accuracy of precision machine tools, Wang et al. [16] promulgated a machine tool error model based on helix theory, identified the machine tool’s weak points, and then optimised its design with harmonic response analysis to assess the device’s layout form. The dynamic properties of machine tools greatly influence the precision of a device’s machining [17]. Lei et al. [18] exploited measurements to determine weak subsystems while enhancing the machine’s settings for operation to increase machine effectiveness. By implementing a dynamic model based on the hypothesis of multibody systems, Ding et al. [19,20] investigated the impact of the spindle and tool’s dynamic properties on the manufactured surface’s midfrequency ripple magnitude. Chen et al. [21] hold that the spindle is just like the main component of the precision device, and the spindle motion accuracy will directly affect the face shape accuracy of the machined surface. In the study of surface quality in fly-cutting machining, the diamond precision fly-cutting process is intermittent, and tool wear and deflection affect the quality of the machined surface and thus the evaluation of surface roughness factors [22,23,24,25,26]. Li et al. [27] conducted tool tip position response experiments by modelling the tip position dynamics to predict the roughness of the machined surface. Bai et al. [28] adopted a neural network based on the ResNet-50 architecture for deep learning to forecast machined surface roughness. Tan et al. [29] used experimental research to determine the influence law of various process parameters on the machined surface roughness and confirmed the ideal process parameters. Cao et al. [30] constructed a theoretical surface morphology model by combining the cutting path and working condition parameters. They confirmed it through examinations to discover the quantitative link between the condition parameters on the surface roughness of the workpiece. The present presents the development of a diamond precision fly-cutting machine for ceramic-copper substrates to increase the productivity and machining accuracy of copper cladding surfaces on ceramic-copper substrates.

The overall layout scheme was determined by performing a harmonic response analysis of the machine’s layout form using finite element analysis, following the study of the structural model of the ceramic-copper substrate and the principle of the diamond fly-cutting machining process. The residual height model of the workpiece surface was established based on the tool tip trajectory equation to investigate the impact of spindle runout error on the machined surface roughness. A multistation design was employed to increases the efficiency of the manufacturing line; the device’s stability was improved through selection of the best design parameters, and parametric variables were employed for determining the critical dimensions of the beam-column structure. The fly-cutter disc assembly was subjected to static and transient analyses to confirm its stability. Lastly, the machine assembly was concluded through a checking of the precision of the Y and Z guides, and the tool’s operation was ensured via fly-cutting machining trials.

## 2. Design Principle of the Diamond Precision Fly-Cutting Device

### 2.1. Ceramic-Copper Substrate Composition

The ceramic-copper substrate structure consists of a ceramic substrate, copper cladding, and photographic film, as shown in Figure 1. One of the copper claddings, through the electroplating process, is coated with copper on the surface of the ceramic substrate due to the effect of the electroplating process causing uneven thickness of the copper cladding surface phenomenon. However, the quality of the copper cladding surface directly impacts the service life of the substrate and, therefore, needs to be in the electroplating after the copper cladding surface processing. In order to ensure the performance of ceramic-copper substrates, it is usually necessary to control the surface accuracy of the copper cladding processing to 15 nm.

### 2.2. Fly-Cutting Processing Method

There are two typical precision diamond fly-cutting machining (or fly-cutting machining) methods related to the installation arrangement of the diamond tool and the machining requirements. Figure 2 displays the two modes of operation: Figure 2a shows the flat surface machining, where the diamond tools are positioned axially along the spindle at the lower end of the fly cutter disc, and Figure 2b shows the free-form surface machining, where the diamond tools are positioned radially along the spindle at the side of the fly-cutter disc.

For different types of workpiece clamping, there are two types of spindle layout: vertical and horizontal. To increase the machining efficiency of the fly-cutting method, the device laid out in this research only needs to remove material from the copper-clad surface of the ceramic-copper substrate, with consideration to the workpiece’s processing requirements and the clamping method Figure 2a.

The machining schematic depicted in Figure 3 shows the fly-cutter disc connected to the spindle along the z-axis vertically feeding to the cutting height; the fly-cutter disc is at the lower end of the installation of counterweights to adjust the dynamic balance in the fly-cutter disc, and the workpiece is placed by the vacuum fixture adsorption, along the horizontal direction of the y-axis for uniform speed feeding.

### 2.3. Fly-Cutting Principle and Accuracy Analysis

Fly-cutting principle and accuracy analysis Figure 4 indicates a type of noncontinuous machining called “flying-cut machining”. During machining, the workpiece is fed uniformly along the horizontal plane while the spindle swings the diamond cutter on the fly-cutter disc at a high speed. The blade only comes in contact with the workpiece once.

By examining the motion trajectory of the tooltip on the flying tool disc and using the previously discussed analysis of the motion of the machining surface and the flying tool disc, we can determine the three-dimensional trajectory equation of the tip motion concerning the tooltip and the coordinate system of the workpiece surface as follows:(1)$$\begin{array}{cc} \; & x\left(t\right)=R\cos\left(2\pi nt\right)\\ \; & y\left(t\right)=R\sin\left(2\pi nt\right)+ft\\ \; & z\left(t\right)=A\left[1+\varepsilon \sin\left(2\pi nt\right)\right] \end{array}$$
where R represents the radius of the fly-cutting disc (mm), f represents the feed speed (mm/s), n represents the spindle speed of spindle rotation (r/min), t is the fly-cutting machining time (s), A is the removal volume (mm), and ε is the runout error of the spindle.

As shown in Figure 5, the axial runout error of the spindle in the form of sinusoidal waves during fluctuating causes the fly-cutting disc to deflect, affecting the tooltip’s relative position on the workpiece’s surface.
(2)$$E_{\varepsilon }=\varepsilon \sin\left(\omega t\right)\;$$
where: $E_{\varepsilon }$ is the runout error of the flying tool disc, and ω implies the angular frequency of spindle rotation (rad/s). This can be expressed as follows:

During fly-cutting, the shape of the material’s surface removed is related to the tool’s geometry through the proximity between the tool and the workpiece, as shown in Figure 4. As a noncontinuous machining process, fly-cutting can be viewed as a motion with two degrees of freedom during the surface removal process: the workpiece’s horizontal feed motion and the spindle’s rotary motion. The residual height is the amount of uncut material on the workpiece’s surface between every two cuts. The theoretical maximum material residual height appears in Figure 6 below [31].
(3)$$R_{max}=r-\sqrt{r^{2}-{\left(\frac{f}{2}\right)}^{2}}$$

The current theory states that the maximum material residual height can respond to the level and trend of the theoretical value of surface roughness to some extent. As a result, the material residual height that results can be considered the workpiece’s theoretical roughness value after the fly-cutting process. The theoretical surface roughness Raof the workpiece can be known as ≈$R_{max}/4\;$ [32], so the theoretical surface roughness is related to the academic maximum material residual height.

Considering the spindle runout error, the machining surface was initially discretised as an X–Y plane comprised of nodes. The N(x,y) coordinates for any node may count as appointed, as illustrated in the figure: position any node N(x,y) at the machining surface of the workpiece.
(4)$$T\left(K\right)=k\cdot dt-\frac{1}{2}dt\cdot \arcsin\left(\frac{y}{r}\right)/\pi$$

Its actual maximum residual height at any node N can be expressed as follows:(5)$${R^{\prime }}_{max}=r-\sqrt{r^{2}-s^{2}}+E_{\varepsilon }$$
(6)$$s=x-\sqrt{r^{2}-y^{2}}+r-f\cdot T\left(K\right)$$
(7)$${R^{\prime }}_{max}=\frac{r-\sqrt{r^{2}-{\left(x-\sqrt{r^{2}-y^{2}}+r-f\cdot \left(k\cdot dt-\frac{1}{2}dt\cdot \arcsin\left(\frac{y}{r}\right)/\pi \;\right)\right)}^{2}}+\varepsilon \sin\left(\omega t\right)}{4}$$
where T (K) is the machining time at node N relative to the Kth tool tip motion trajectory (s), r is the radius of the tool’s arc (mm), f is the workpiece’s horizontal feed rate (mm/s), s is the horizontal distance between node N(x,y) and the tool tip trajectory at any place (mm), and ${R^{\prime }}_{max}$ is the roughness inclusive of the runout error of the spindle.

Equation (Equation 7), as can be observed, shows a relationship between the spindle’s runout error and the roughness of the workpiece’s actual machined surface. In the machining process, the solid surface roughness results from the superposition of various factors, in which the spindle will produce sinusoidal periodic runout error, affecting the surface accuracy of the machined surface. Therefore, it will cause the tool to generate an offset caused by the inconsistency between the theoretical machining surface and the actual machining surface. Consequently, it is necessary to consider analysing the influence of spindle runout error on fly-cutting.

## 3. Design of the Ceramic-Copper Substrate Fly-Cutting Device

### 3.1. Form of the Installation’s General Layout

In the design of precision devices, the layout form of the device will affect the static and dynamic characteristics of the device itself, as well as thermal deformation, which directly affects the performance of the device, so the layout form of the precision device plays a crucial role in the machining quality of the machined surface. The common precision device layout form has a gantry layout and cantilever layout of these two forms.

A gantry layout frequently occurs in the form of a symmetrical beam-column structure, which has good static and dynamic characteristics and can effectively reduce the impact of thermal deformation of the machine tool. A cantilever layout typically includes an open-column structure, which has the advantage being compact and easy to assemble, but its resistance to thermal deformation is poor, and there is considerable flexural deformation. The two layout schemes’ gantry layout and the cantilever layout were comparedto guarantee the performance of the fly-cutting machine designed in this study. The harmonic response results of the modal and harmonic response analysis of these two layout schemes are displayed in Figure 7, where the first-order intrinsic frequencies of the cantilever layout form and the gantry layout form are 245 Hz and 346 Hz, respectively. In the range of 0–1000 Hz, there are three obvious resonances of the cantilever layout form and two apparent resonances of the gantry layout form. The gantry layout form has good harmonic response properties. Given that the gantry layout form exhibited favourable dynamic properties, the gantry layout was selected as the machine tool’s layout scheme in this study to enhance the designed device’s performance.

Based on the above finite element analysis results, the overall layout was then designed after the symmetrical gantry layout was determined to be the layout form of the device developed in this study. A marble base, bed, column, beam, electro spindle, flying cutter disc, pneumatic fixture, x/y-axis skateboard, and others were the main components of the device designed in this study, depicted in Figure 8. The main body of the device described in this paper was made of concrete and vibration-resistant marble to improve the stability of the device.

### 3.2. Design Indicators for Machine Tools

Requirements for the designed device in the fly-cutting processing were a machining surface roughness Sa better than 15 nm and a parallelism of the guideway better than 4 μm/800 mm. This was assumed to ensure the machining performance of the machine tool as well as to improve the machining efficiency of the ceramic-copper substrate machining surface according to the machining requirements for developing the device design indicators shown in Table 1.

### 3.3. Structural Design of the Beam-Column Arrangement

To achieve rapid thinning of the ceramic-copper substrate copper cladding surface and improve production efficiency, the fly-cutting device designed in this study employed a double-station design, which includes two stations symmetrically arranged on the same marble beam, which serves to semifinish and finish the ceramic-copper substrate machining surface, respectively. Within the crossbeam guide mounting surface, a remote force of 150 N was applied. The single-station and dual-station schemes were verified by employing Workbench. Figure 9 displays the results of the analysis. The fly-cutting device with a dual-station design had a maximum deformation at the crossbeam of 0.16488 μm, which was less than the maximum deformation of the single-station design. Consequently, the maximum deformation may be made by using the dual-station design. Thus, while adhering to the design index, the double-station design can increase production efficiency. Consequently, the two-station design scheme appears within this paper.

After the overall structure of the device was determined, the details of the device needed to be optimised and designed. The main body of the device included a symmetrical gantry beam-column structure. To further improve the static and dynamic performance of the device, the positioning and setting dimensions in the beam-column structure were parametrically varied in the structure design. As shown in Figure 10, the distance between the setting-up column and the two ends of the beam is a, the width of the column is b, the span is c, and the total length of the beam is d. The machining object of the device is a ceramic-copper substrate with a size of 190 × 140 mm, and to ensurethe Y and Z sliding plate, the vacuum jig, and the spindle’s mounting and workspace, the dimensions of the device’s column structure were determined to be 300 × 220 mm (c × g).

The equations for the other parameters in the beam-column structure are as follows:(8)$$\begin{array}{cc} d-2 & b-2a=300\\ e & -f=220\\ 0 & \leq a\leq 40\\ 60 & \leq b\leq 80\\ 420 & \leq d\leq 540\\ 420 & \leq e\leq 440\\ 200 & \leq f\leq 220 \end{array}$$

The beam-column structure’s parametric structural design was determined according to the methodology described above, and with the displacement of the column from the end face of the beam as the variable, finite element analysis was carried out, with the ideal design parameters derived based on the optimal simulation results. Table 2 displays the design parameters both before and after optimisation.

To consider the effect gravity, a remote force of 150 N was exerted at the Z-rail mounting surface of the crossbeam, and finite element analysis was carried out on the crossbeam-column structure under the two materials of marble and structural steel by using Workbench software 2020R2. The simulation results appear in Figure 11, which clearly shows that selecting marble as the cross-beam column structure can effectively reduce the beams’ static deformation. Hence, a marble crossbeam-column structure was chosen.

Marble was chosen as the material for the crossbeam-column structure to lessen the crossbeam’s distortion. In Figure 12, the machine’s harmonic response curve before the optimisation is represented by the blue dashed line, while the red solid line represents the machine’s harmonic response curve after optimisation. The machine’s first-order intrinsic frequency increased from 405 to 420 Hz when compared to the preoptimisation parameters, and the maximum deformation of the Z-directional orientation of the marble crossbeam of the device slumped from 0.55433 μm to 0.063699 μm. Thus, optimising the beam-column structure utilising parametrised variables can enhance the device’s static and dynamic performance.

### 3.4. Structural Design of the Flying-Cutting Disc Components

The fly-cutting approach requires that the flying-disc components mounted on the diamond tool rotate the workpiece surface continuously at high speed. As a result, the flying-disc components must have good dynamic and static properties. A fly-cutting device used at the end of the fly-cutting processing mode was designed in this study, with the diamond tool migrating to the fly-cutter disc side.The influence of spindle runout on the tool path was reduced by using a tapered adapter to connect the spindle to the flying tool disc. The fly-cutter disc components included a symmetrical structure design to facilitate dynamic balance debugging of the fly-cutter disc.The results above indicate that the fly-cutting disc’s overall layout performs as intended. Figure 13 depicts the overall configuration of the fly-cutter disc described in this paper, which includes the counterweight block, diamond tool, electric spindle, and fly-cutter disc.

As the fly-cutting disc has a high linear speed, which directly affects the machining quality, to ensure the flying cutter’s processing stability, two schemes were implemented to analyse the fly-cutter disc components via Workbench software. To account for gravity, the rotary support constraints on the fly cutter disc were set to a rotational speed of 800 r/min, and the overall structure of the fly-cutter disc of structural steel material was subjected to finite element analysis.As shown in Figure 14, the b scheme had a static self-weight maximum deformation of 0.16736 μm and a transient maximum deformation of 0.16772 μm; the use b scheme’s stability was better than that of the a scheme Based on the above analysis, to guarantee the device’s design index, the b scheme was chosen as the design scheme of the flying knife disc.

## 4. Measurement and Verification Experiments

### 4.1. Complete Machine Assembly

After the design and processing were concluded, the machine tool created in this study needed to be constructed and debugged. The machine tool made for this paper, as illustrated in Figure 15, has two machining stations that allow for semifinishing and finishing. The holding motor drives the screw-nut sub of the Z-axis to produce the high-precision linear motion of the Z-direction skid; the axes S1 and S2 originate from the electric spindle to obtain a higher rotational speed to remove surface materials rapidly. The fly-cutter disc spindles S1 and S2 are positioned within the Z-direction skids corresponding to the two stations. A pair of nuts serve for anchoring the Y-skate at the bottom, and a servomotor rotates the screw to provide the skate’s linear motion. Spindles drive axes S1 and S2 to achieve high speeds and quick removal of surface materials. To ensure the precision of the Z-direction movement, the spindles go the axes S1 and S2 to obtain high speeds and achieve fast removal of surface materials. The servomotor works to rotate the screw, enabling the Y-skate to move linearly. The nut pair sits at the bottom of the skate. To ensure the positioning accuracy of the Z-direction movement of the Z-direction skateboard during fly-cutting, the Z-axis adopts a linear scale with a resolution of 0.05 μm, which realises the high-precision positioning of the Z-direction skateboard.

### 4.2. Measurement of Guideway Straightness

In the fly-cutting process, the Z- and Y-direction guides must move in a straight line, which has an essential influence on the accuracy of the machined surface. As a result, measuring and verifying the precision of the Y and Z guide rails is required to increase the accuracy of fly-cutting machining. The data from the Matlab fitting curve were collected, with a micrometre measurement through the Y-axis guide being employed, along with a rail measurement range of 800 mm. The skateboard is moved at a constant speed of 20 mm/min along the Y-axis from the start to the end of the movement. Figure 16 depicts the tilt-removal principle model. The tilt slope of the fitted picture curve, *k*_1_, can be written as follows:(9)$$z_{1}\left(i\right)=k_{1}\times x\left(i\right)$$
(10)$$x\left(i\right)=L+v_{x}\cdot t$$
(11)$$z^{\prime }\left(i\right)=z\left(i\right)-z_{1}\left(i\right)$$
(12)$$z^{\prime }\left(i\right)=z\left(i\right)-\frac{\sum_{i=1}^{n}x\left(i\right)z\left(i\right)-\sum_{i=1}^{n}x\left(i\right)z\left(i\right)/n\;}{\sum_{i=1}^{n}x^{2}\left(i\right)-\sum_{i=1}^{n}x^{2}\left(i\right)/n\;}x\left(i\right)\;$$
where $z\left(i\right)$ is the measurement of the guideway vice parallelism data, $x\left(i\right)$ is the measurement of real-time X directions, n is the measurement of the number of acquisition points, $z_{1}\left(i\right)$ is the guideway vice-tilt measurement of the quantity of error, $z^{\prime }\left(i\right)$ is the principle of removing the tilt error in the processing of the data, $v_{x}$ is the data measurement and acquisition of the speed of the movement, and t is the detailed measurement and acquisition of time.

As illustrated in Figure 17, the marbling size was utilised as the standard for guide rail straightness measurement. The surface face shape accuracy on the scale is less than λ/10. The least squares method was used to eliminate the degree of tilt in the data to accurately obtain the straightness data of the rail sub.

The formula utilises the collected $z\left(i\right)$ data to obtain the $z^{\prime }\left(i\right)$ data through Matlab fitting; in Figure 18, the red line represents the conclusion of fitting the measurement data via removal of the tilt error amount of the guideway vice tilt measurement, and the blue line indicates the initial data fitting image after measurement; the fitted image demonstrates that the X-axis motion straightness error is 2.42 μm/800 mm, which satisfies the design indexes.

The accuracy measurement results of the same approach to the Z-way guide are shown in Figure 18; the Z-axis motion straightness error is 2.32 μm/200 mm, thus meeting the design indicators.

### 4.3. Fly Cutting Processing Experiment

After the assembly of the device designed in this study and completion of the accuracy measurements, the processing accuracy of the device was verified by carrying out experiments and measurements on the fly-cutting machining of a 190 mm × 140 mm ceramic-copper substrate. Two vacuum cleaners were installed at each station to prevent the loss of waste debris on the machined surface during the removal process, which could affect the machined surface’s machining quality.

After being sucked through a vacuum fixture, the machined components were used in fly-cutting experiments. The fly-cutting machining experimental parameters included a 0.1 mm cutting amount, an 800 rpm cutting spindle speed, a 100 mm/min workpiece feed speed, an arc radius of 4791.27 μm, a front angle of 0°, a back angle of 10°, and a diamond tool height of 6.189 mm, as shown in Figure 19.

The roughness of the ceramic-copper substrate’s copper-coated surface, which is ignored by fly-cutting, was measured with the ZYGO white light interferometer to be Sa133.67 nm, as shown in Figure 20. After the sample was fly cut via the device proposed in this paper, in order to meet the device’s design parameters, the surface roughness was measured as Sa9.058 nm.

## 5. Conclusions

We created a precision diamond fly-cutting tool that was applied specifically to the ceramic-copper substrate to increase the machining accuracy and productivity of the substrate’s machining surface.

(1) By analysing the working principle of fly-cutting processing and considering the spindle movement accuracy, we constructed a roughness prediction model based on the tool tip trajectory, and it appeared that the runout error of the spindle led to the change of the tool tip trajectory, which directly affected the machining surface roughness.

(2) Through an harmonic response analysis and comparison results, the gantry symmetric structure was chosen as the layout form of the device to improve the stability of the machine. The use of a double-station design was employed to enhance productivity, and the overall layout of the critical dimensions underwent parametric variable design. Static, modal, and harmonic response analyses were used to optimize the general layout of the device and the fly-cutting disc components. The optimised simulation results showed that after the optimisation of the device’s first-order, the first-order intrinsic frequency of the device was increased to 420 Hz, and the maximum deformation of the crossbeam’s Z-direction was reduced to 0.063699 μm; the maximum deformation of the flying-cutter disc component’s static deadweight was 0.16736 μm, and the maximum deformation of the transient state was 0.16772 μm, which could effectively improve the machine tool’s static and dynamic characteristics.

(3) The LK-G5000 laser sensor was used for measuring the precision of the diamond fly-cutting device. The measurement results showed that the Y-axis guideway straightness was better than 2.42 μm/800 mm and that the Y-axis guideway straightness was better than 2.32 μm/200 mm, which met the design indicators.

(4) Fly-cutting processing experiments and measurements were conducted on the developed precision diamond fly-cutting device with 190 mm × 140 mm ceramic-copper substrate.The measurement of the surface roughness of the processed surface Sa was 9.058 nm, which met the device’s design specifications.

This paper mainly discusses the complete design of a precision fly-cutting machine tool and the use of spindle runout error to examine its impact on machining surface roughness. As the design of the fly-cutting machine also needs to account for manufacturing costs, the online detection of tool wear, compensation, and other issues, a more in-depth understanding of the fly-cutting process is needed. To this end, further experimental follow-up should be conducted.

## Figures and Tables

**Figure 1 materials-17-01111-f001:**
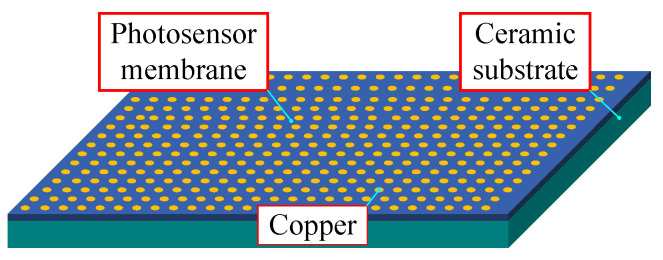
Ceramic-Copper Substrate Composition Mechanism.

**Figure 2 materials-17-01111-f002:**
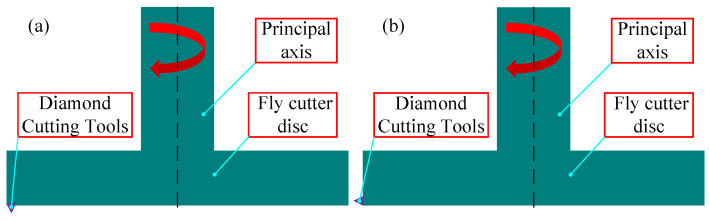
Diamond fly-cutting processing method. (**a**) End fly-cutting. (**b**) Radial fly-cutting.

**Figure 3 materials-17-01111-f003:**
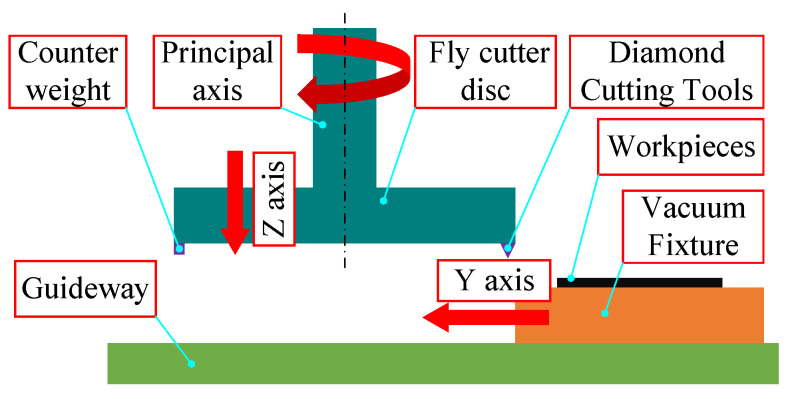
Schematic diagram of the fly-cutting process.

**Figure 4 materials-17-01111-f004:**
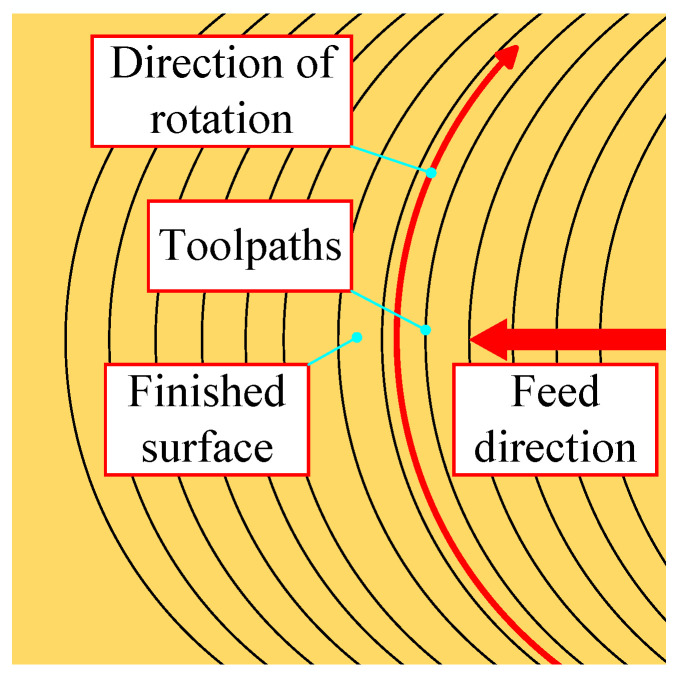
Fly cutting trajectory diagram.

**Figure 5 materials-17-01111-f005:**
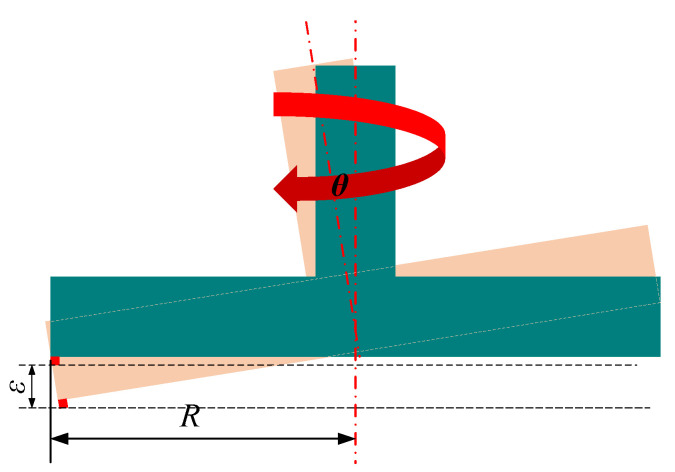
Schematic diagram of fly-cutting process.

**Figure 6 materials-17-01111-f006:**
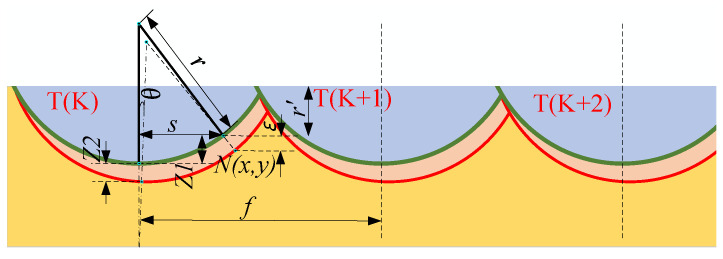
Residual height model after the fly-cutting process.

**Figure 7 materials-17-01111-f007:**
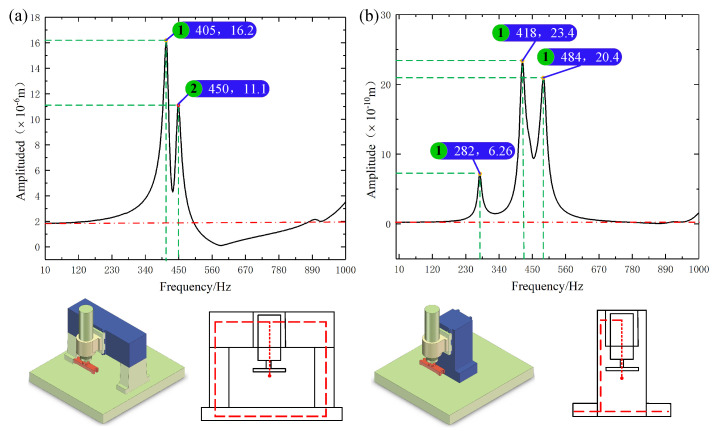
Comparison of the harmonic response analysis of the two schemes. (**a**) Gantry structure. (**b**) Cantilever layout.

**Figure 8 materials-17-01111-f008:**
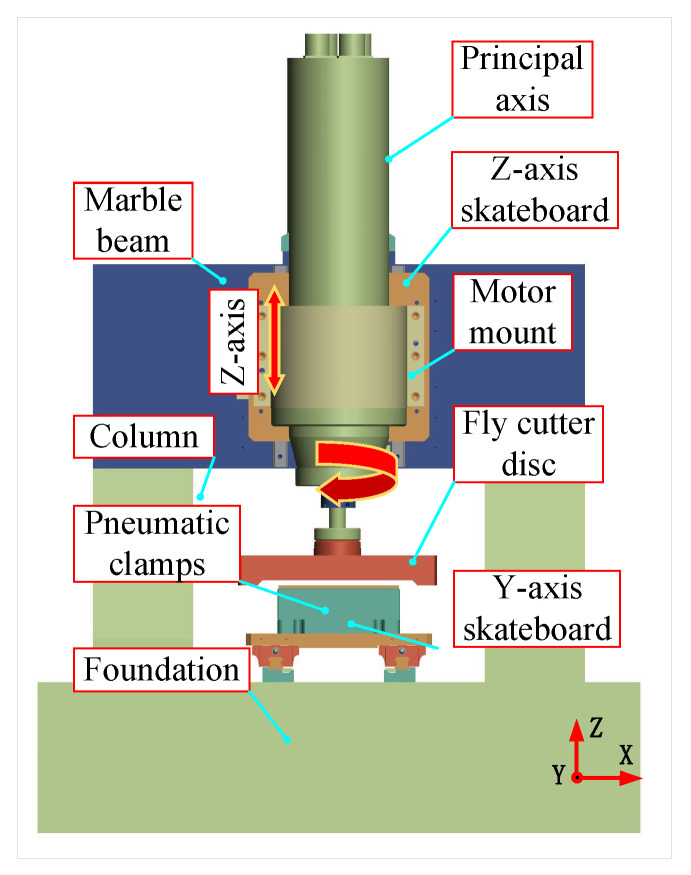
The general structure of the device.

**Figure 9 materials-17-01111-f009:**
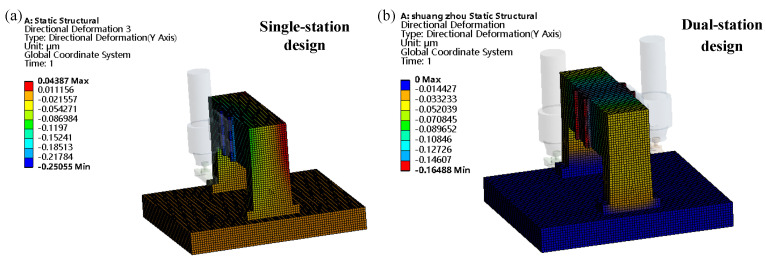
The general structure of the device. (**a**) Single station. (**b**) Dual station.

**Figure 10 materials-17-01111-f010:**
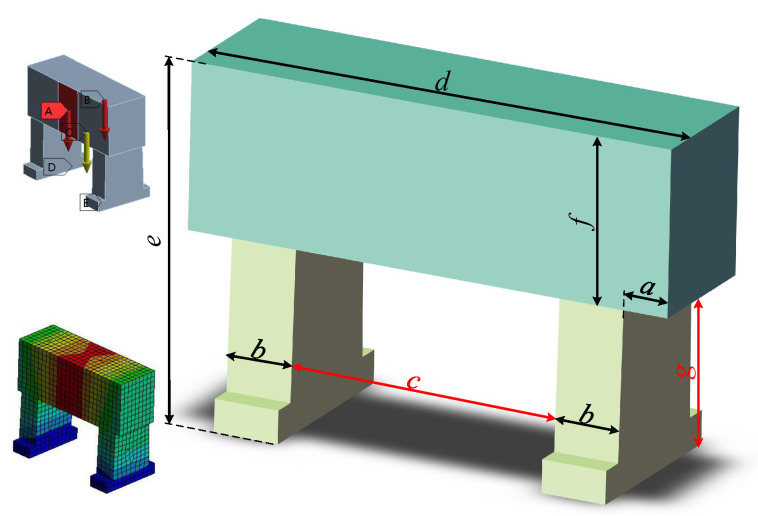
Parametric variable in the design of the beam-column structure.

**Figure 11 materials-17-01111-f011:**
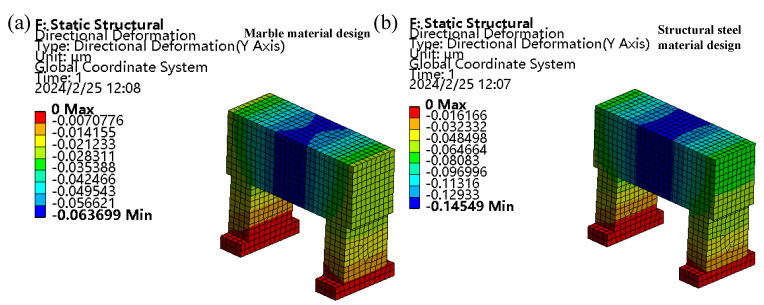
Static deformation of the beam column. (**a**) Marble material design. (**b**) Structural steel material design.

**Figure 12 materials-17-01111-f012:**
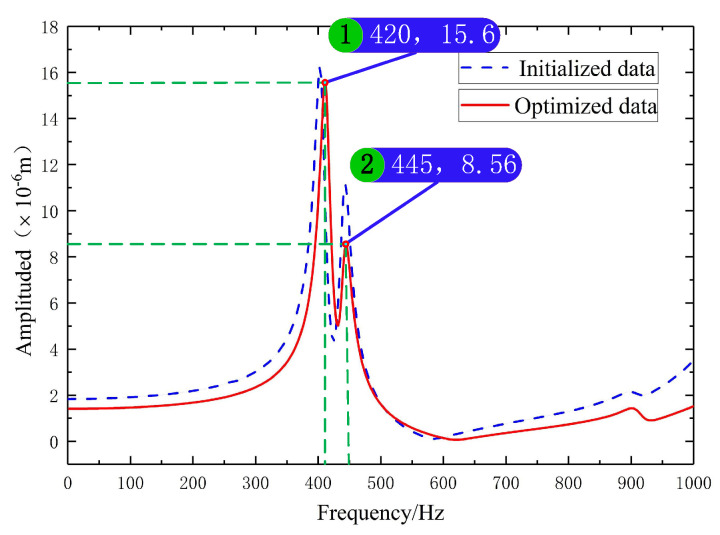
Optimised machine harmonic response.

**Figure 13 materials-17-01111-f013:**
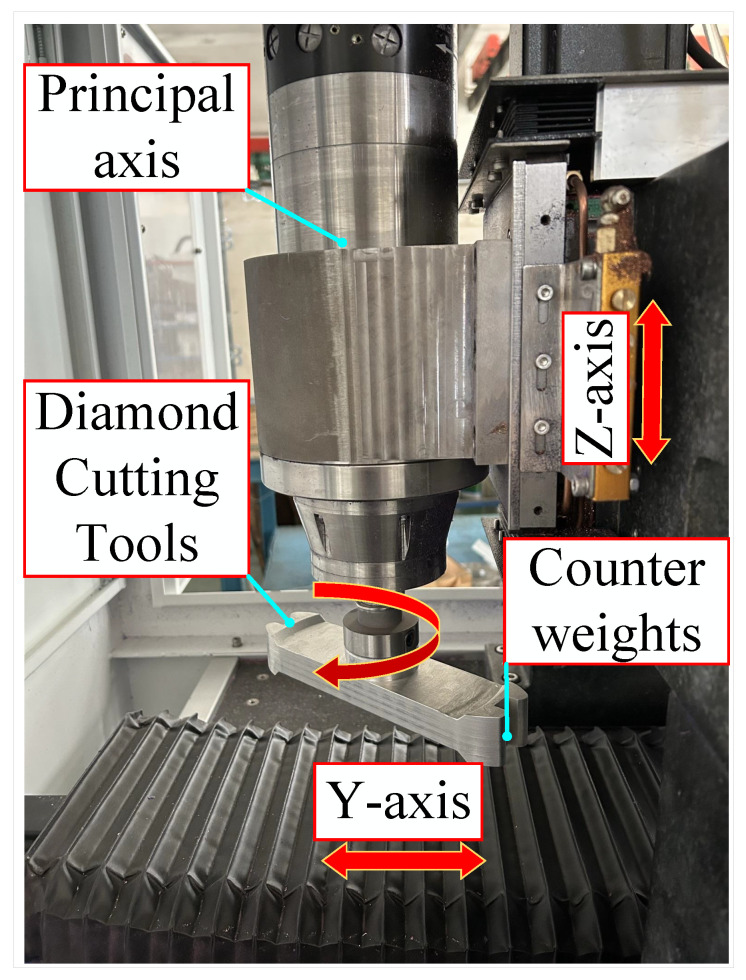
Fly-cutter disc component structure.

**Figure 14 materials-17-01111-f014:**
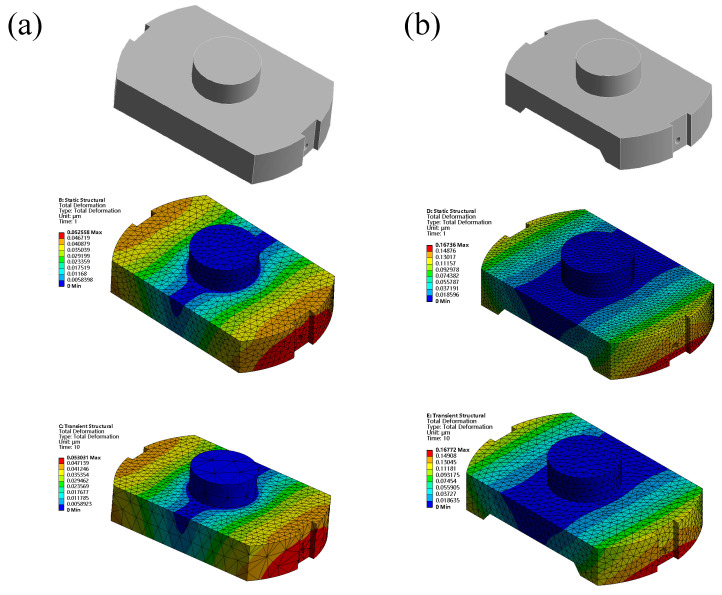
Comparison of the fly- cutter disc options. (**a**) Original programme. (**b**) Lightweight programme.

**Figure 15 materials-17-01111-f015:**
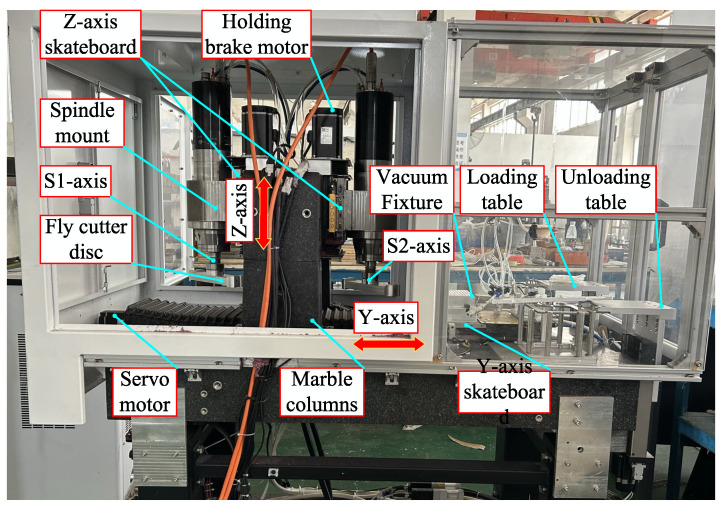
Complete machine assembly.

**Figure 16 materials-17-01111-f016:**
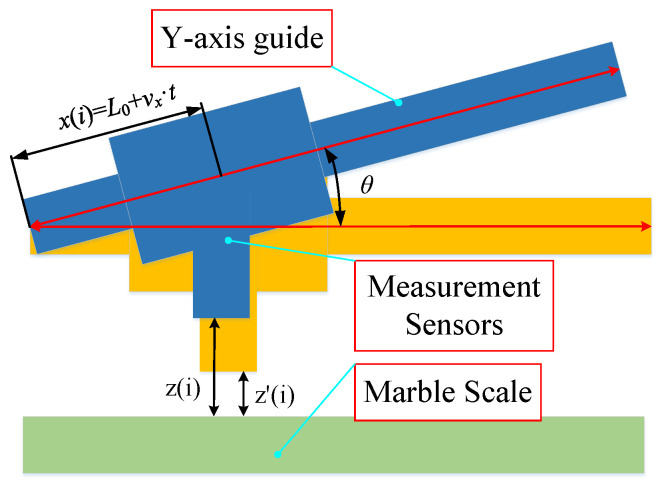
Corrected model for guideway straightness skew.

**Figure 17 materials-17-01111-f017:**
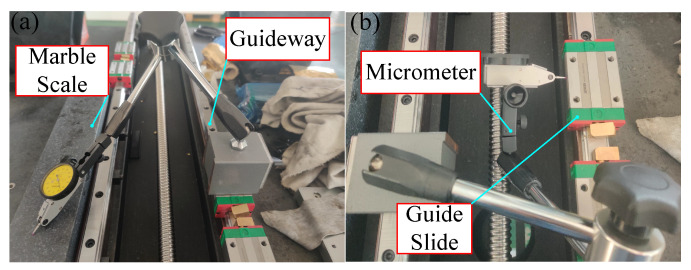
Guideway accuracy Measurement. (**a**) Straightness measurement. (**b**) Flatness measurement.

**Figure 18 materials-17-01111-f018:**
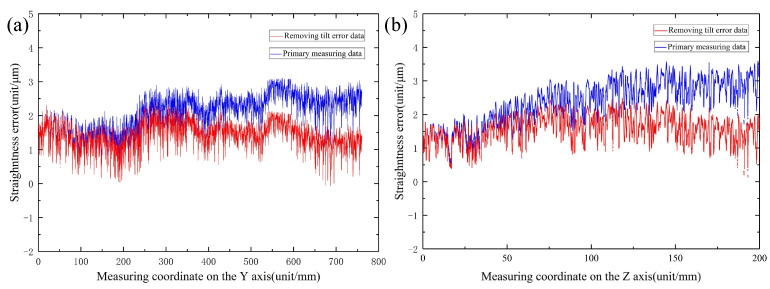
Guideway straightness measurement results. (**a**) Y-axis guide measurement results. (**b**) Z-axis guide measurement results.

**Figure 19 materials-17-01111-f019:**
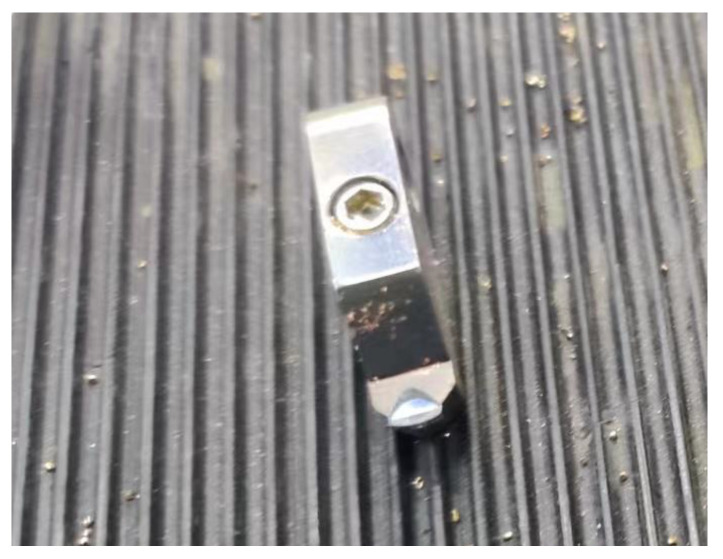
Diamond cutting tools.

**Figure 20 materials-17-01111-f020:**
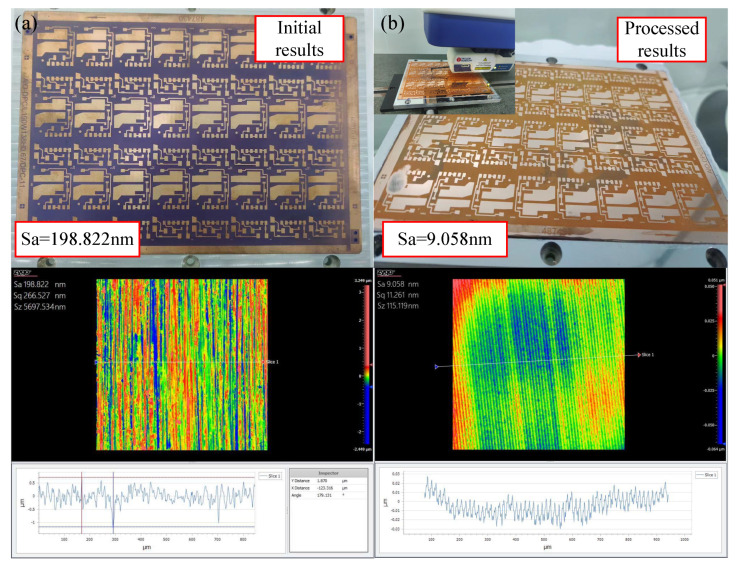
Roughness measurement results. (**a**) Initial results. (**b**) Processed results.

**Table 1 materials-17-01111-t001:** Design index of precision fly-cutting device.

Typology	Fly-Cutting
Distance ravelled	Y-axis: 800 mm Z-axis: 5 mm
Guideway straightness error	Y-axis: 4 μm/800 mm Z-axis: 3 μm/200 mm
Minimum feed speed	Y-axis: 10 μm/s Z-axis: 1 μm/s
Spindle runout error	Spindle: 0.5μm
Workpiece material	Copper
Spindle speed	0–1000 rpm
Size of workpiec	190mm×140mm
Workpiece roughness	Sa ≤15nm

**Table 2 materials-17-01111-t002:** Parameters before and after optimisation of the crossbeam-column structure.

Parametric	a	b	c	d	e	f	g	Frequency
Initial size (mm)	0	75	300	450	440	220	220	405 Hz
Optimised size (mm)	20	80	300	500	420	200	220	420 Hz

## Data Availability

Data are contained within the article.

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
