# Peer review of "Design and Study of Machine Tools for the Fly-Cutting of Ceramic-Copper Substrates"

_materials, 2024, doi:10.3390/ma17051111_

Round 1
Reviewer 1 Report
Comments and Suggestions for Authors
Please refer to the reviewer report attached.

The authors should check their manuscript for grammar and spelling to ensure that their work is error-free.
Author Response
|
Point-by-point response to Comments and Suggestions for Authors |
|
Comments 1:The authors should mention their novel aspects of research as well as the benefits ofimplementing their machine tool against those reposted by the current industrial practices andtrends. |
|
Response 1: Thank you very much for your valuable advice, now the factory for the treatment of copper cladding for the use of abrasive belt grinding process its processing accuracy is low and inefficient; we use the fly cutting process on the surface of the copper cladding layer processing, can improve the surface quality of the copper cladding layer to a large extent. Some of the corrections have been made in the revised version and have been highlighted in blue. It can be found on page 1, second paragraph, line 25. |
|
Comments 2: The authors should mention the standards necessary for designing, developing and finallyimplementing - using such a machine tool. The aspects and technical attributes found instandards should clearly mention the requirements of precision, machine tool vibration errorsrigidity issues, stability indicators etc, The authors should relate their concepts to such standardsand guidelines. |
|
Response 2: Thank you for some advice on this paper on precision machine tool design, this paper is designed by the precision fly cutting machine tool is the first according to the actual processing requirements of the design indicators, its design indicators mainly include the spindle runout error, the straightness accuracy of the guideway, and through the machining test, the surface roughness of the machined sample parts to verify the design accuracy of the machine tool. For the actual measurement of machine vibration because of equipment and financial constraints, and in order to be able to collect the relevant test data. More in-depth research will be carried out in the subsequent iteration of machine tool upgrading. |
|
Comments 3:The authors should mention the cost of building such as machine tool for fly-cutting of ceramiccopper substrates and refer to the standard requirements of such materials in terms of qualityand surface finish. |
|
Response 3: For this paper, the design of the fly-cutting machine tool is mainly for the ceramic - copper substrate copper cladding surface processing, in order to be able to successfully complete the subsequent processes of the substrate, generally only need to require the substrate's surface roughness is better than 15nm, so in the fly-cutting machine tool design index table 1 proposed requirements for the surface roughness of the workpiece is better than 15nm. |
|
Comments 4:The authors mention the very important attribute of run-out error in spindle. They should referalso to the systems and technology accompanying the tool design for maintaining the error atlow levels. For example they should mention systems such as conical adapters or collets to safelyclamp the different tools and ensure low run-out. |
|
Response 4: Thank you for adding to the shortcomings in the design of this article, which has been improved based on your suggestions. Some of the corrections have been made in the revised version and have been highlighted in blue. It can be found on page 9, second paragraph, line 274-276. |
|
Comments 5:The authors should make also comments concerning the CNC control of their development andon-line monitoring of tool wear, compensation issues, cutting force components, etc. |
|
Response 5:As the fly-cutting machine designed in this paper needs to control the manufacturing cost, for the online detection of tool wear, compensation and other issues in this paper and to make a more in-depth understanding,and explained in the discussion section of the article. Some of the corrections have been made in the revised version and have been highlighted in blue. It can be found on page 13, line 398-404. |
|
Comments 6:Some figures involve information that with the current fonts and size cannot be easily reviewed.The authors should enlarge the fonts of such figures to be noticeable and well-presented. |
|
Response 6:Thank you very much for your careful review, changes have been made to the images. |
|
Comments 7:The authors should refer to the limitations of their current work and how these may beovercome by proposing future research and new directions related to their problem statement. |
|
Response 7:We appreciate your suggestions for articles and have added them. Some of the corrections have been made in the revised version and have been highlighted in blue. It can be found on page 13, line 398-404. |
|
4. Response to Comments on the Quality of English Language |
|
Point 1:The authors should check their manuscript for grammar and spelling to ensure that their work is error-free. |
|
Response 1:We made the appropriate changes to the language of the article as requested by the expert assessors. |

Reviewer 2 Report
Comments and Suggestions for Authors
The main question addressed by the research is the design and study of machine tools for fly-cutting ceramic-copper substrates. The research aims to improve the machining accuracy and productivity of the machining surface of ceramic-copper substrates by developing a precision diamond fly-cutting tool.
In terms of originality and relevance, the research addresses a specific gap in the field related to the machining of ceramic-copper substrates, which are important components in various high-power applications such as new energy vehicles, electric locomotives, high-energy lasers, and integrated circuits. The use of precision diamond fly-cutting processing machine tools is introduced to overcome the limitations of existing abrasive strip polishing procedures, providing a more efficient and accurate solution.
The research is relevant as it contributes to the advancement of machining technology for ceramic-copper substrates, with a focus on improving surface quality. The study involves an in-depth analysis of the fly-cutting machining principle, the structural makeup of the substrate, and the impact of spindle runout error on machined surface roughness. The research also includes structural optimization design using finite element software, modal and harmonic response analysis, and parametric variable optimization to enhance machine stability and motion accuracy.
Overall, the research is both original and relevant, addressing a specific need in the field of machining for ceramic-copper substrates and presenting a comprehensive approach to designing and optimizing precision diamond fly-cutting tools for improved accuracy and productivity.
The references are numerous and relevant.
The figures are good quality.
The equations are well written, and easy to follow.
I have detected only a few typos:
Line 109: "flying cut machining." --> "flying cut machining". (The end of the sentence should be outside of the quotation mark.
Figure 7 caption: “Omparison” --> “Comparison”
Line 214: Why is there a line break after the word “Structural”?
Line 350: “2.32mm200mm” --> “2.32mm/200mm” (should be similar to the other such errors, e.g. as in Lines 346, 390, 391.
Line 386: “-7” should be in upper index (similar to Lines 384-385).
I think these typos should be corrected, after that the manuscript can be accepted for publication.
Author Response
Dear reviewers:
Greetings!
Thank you very much to the reviewers for your comments on this paper. The questions you have raised about this paper are the key difficulties encountered in the research of this paper. The following will respond to your questions one by one.
|
Point-by-point response to Comments and Suggestions for Authors |
|
Comments 1: Line 109: "flying cut machining." --> "flying cut machining". (The end of the sentence should be outside of the quotation mark. |
|
Response 1: Thank you very much for correcting the errors in this article, after careful examination of the relevant errors are corrected to " "flying cut machining"." Some of the corrections have been made in the revised version and have been highlighted in blue. It can be found on page 3, third paragraph, line 115. |
|
Comments 2: Figure 7 caption: “Omparison” --> “Comparison” |
|
Response 2: Thank you very much for correcting the errors in this article, after careful examination of the relevant errors are corrected to "Comparison." Some of the corrections have been made in the revised version and have been highlighted in blue. It can be found on page 6, , line 195. |
|
Comments 3:Line 214: Why is there a line break after the word “Structural”? |
|
Response 3: Thank you very much for correcting the error in this article, after careful examination is caused by the right side of the figure eight layout issues. Some of the corrections have been made in the revised version and have been highlighted in blue. It can be found on page 6, , line 217. |
|
Comments 4:Line 350: “2.32mm200mm” --> “2.32mm/200mm” (should be similar to the other such errors, e.g. as in Lines 346, 390, 391. |
|
Response 4: Thank you very much for correcting the errors in this article, we have reviewed it carefully and the relevant errors have been corrected. Some of the corrections in the revised version have been highlighted in blue. See page 12, paragraph 2, line 352. |
|
Comments 5:Line 386: “-7” should be in upper index (similar to Lines 384-385). |
|
Response 5:Thank you very much for correcting the errors in this article, we have reviewed it carefully and the relevant errors have been corrected. Some of the corrections in the revised version have been highlighted in blue. See page 13, paragraph 2, line 387. |
|
4. Response to Comments on the Quality of English Language |
|
Point 1:I am not qualified to assess the quality of English in this paper |
|
Response 1:We made the appropriate changes to the language of the article as requested by the expert assessors. |

Reviewer 3 Report
Comments and Suggestions for Authors
Comment 11 The text of the article is readable, but it would be advisable to adjust the English grammar.
Author Response
Dear reviewers:
Greetings!
Thank you very much to the reviewers for your comments on this paper. The questions you have raised about this paper are the key difficulties encountered in the research of this paper. The following will respond to your questions one by one.
|
Point-by-point response to Comments and Suggestions for Authors |
|
Comments 1:The title, the abstract, the keywords and the text of the contribution are in agreement. |
|
Response 1: Thank you very much for correcting the errors in this paper and have aligned the title, abstract, keywords and body of the manuscript. Some of the corrections have been made in the revised version and have been highlighted in blue. Some of the corrections have been made in the revised version and have been highlighted in blue. It can be found on page 1, line 20. |
|
Comments 2: The literature review is in order within the given topic. |
|
Response 2: Thanks to your suggestions for articles, we have been interested in categorising the literature review according to three themes. These are in terms of design and optimisation of precision machine tools, in terms of research on improving the machining accuracy of precision machine tools, and in terms of research on the surface quality of fly-cutting machining. Some of the corrections have been made in the revised version and have been highlighted in blue. Some of the corrections have been made in the revised version and have been highlighted in blue. It can be found on page 1, line 35; page 2, line 42; page 2, line 54. |
|
Comments 3:Image editing required. Figure 1. Label what the colour blue is. Figure 2. and 3. Complete the axes of rotation and the symbol for rotation. Complement the rocking movement? Add a), and b) to the text of the picture. Figure 4. Complete the rocking movement. Fill in what the black circles mean. Figure 5. and Figure 6. Add symbols to the text of the figure: R, θ, ε, T(K), .... Figure 7., 9., 11., 14., 18., 19. Add a), and b) to the text of the image. Figure 8. Complete the axis and symbol of rotation and rocking motion. Figure 13. add the symbol (mark) of the rocking movement. |
|
Response 3: All photographs have been modified as you have suggested. Fig. 1 labelled in blue is the photographic film. Fig. 2 as well completes the addition of the rotary axis. The black circle in Fig. 4 represents the trajectory of the tip of the knife. The additions to Figures 5 and 6 for the symbols have been completed. Figures 7, 9, 11, 14, 18 and 19 have been supplemented with symbols and comments. |
|
Comments 4:It is appropriate to supplement the image for the shape and geometry of the diamond cutting tool. |
|
Response 4: Diamond tool images have been supplemented according to your suggestions. It can be found on page 12, Figure19. |
|
Comments 5:Line 148 The article is in the spirit of precision machining, and the authors used an equation that approximately calculates the roughness value Ra depending on the value of Rmax. I don't think it's appropriate to state this like that. There are literary sources where Ra for a hemispherical cutting tool can be expressed mathematically, e.g. A New Approach to Calculating the Arithmetical Mean Deviation of a Profile during Copy Milling (svjme.eu/?ns_articles_pdf=/ns_articles/files/ojs3/1076/submission/1076-1-2515-1-2-20171107.pdf&id= 4470) and others. |
|
Response 5:Thank you very much for your expression of roughness Ra in this paper and for reviewing the literature based on your suggestions. According to this article: A New Approach to Calculating the Arithmetical Mean Deviation of a Profile during Copy Milling, which you recommended, for conventional machining can be approximated by Ra to Rt/(3-5), it is possible that in precision machining on roughness also. It may not be appropriate to describe the roughness in precision machining, but this paper mainly focuses on the design of fly-cutting machine tools, and the roughness is described in order to analyse the definition of roughness and to study the impact of the designed machine tools on machining accuracy by introducing the spindle runout error. Finally, we would like to thank you for your suggestion to conduct a more in-depth study on the surface morphology of flycutting in the future.
|
|
Comments 6:Add units for the parameter of all equations. |
|
Response 6:Units have been added to all parameters in the article. Some of the corrections have been made in the revised version and have been highlighted in blue. |
|
Comments 7:For equation (3), list the literature. |
|
Response 7:References for equation (3) are already listed. Some of the corrections have been made in the revised version and have been highlighted in blue.It can be found on page 4. |
|
Comments 8:In Table 1, it is appropriate to add the speed of the vf device in individual axes., and to add the speed of the oscillating movement |
|
Response 8:Thank you for your suggestion, it has been added in Table 1 about the minimum feed speeds of Y and Z axes, which are different in order to control the cost. For the spindle, the speed range is selected as 0-1000rpm. Some of the corrections have been made in the revised version and have been highlighted in blue.It can be found on page 7,Table1. |
|
Comments 9:It is appropriate to state the criterion for device dimension optimisation listed in Table 2. |
|
Response 9:With the displacement of the column from the end face of the beam as the variable, finite element analysis was carried out respectively, and the ideal design parameters were derived based on the optimal simulation results. Some of the corrections have been made in the revised version and have been highlighted in blue.It can be found on page 8,line 247-249. |
|
Comments 10:It is appropriate to provide an image for the roughness of the Sa surface in the 3D view. |
|
Response 10:Thank you for your suggestions, we have added to the pictures. Some of the corrections have been made in the revised version and have been highlighted in blue.It can be found on page 13,Figure20. |
|
Comments 11:It is appropriate to supplement the Discussion chapter. |
|
Response 11:Thank you for adding to the shortcomings in the design of this article, which has been improved based on your suggestions. Some of the corrections have been made in the revised version and have been highlighted in blue. It can be found on page 9, second paragraph, line 274-276. |
|
Comments 12:The conclusion would be appropriate to write in the past tense. |
|
Response 12:Thank you for your suggestion, the tense in the conclusion section has been changed. Some of the corrections have been made in the revised version and have been highlighted in blue. It can be found on page 12, second paragraph, line 372-298. |
|
4. Response to Comments on the Quality of English Language |
|
Point 1:Comment 11 The text of the article is readable, but it would be advisable to adjust the English grammar. |
|
Response 1:We made the appropriate changes to the language of the article as requested by the expert assessors. |
